# Engineered Microphysiological Systems for Testing Effectiveness of Cell-Based Cancer Immunotherapies

**DOI:** 10.3390/cancers14153561

**Published:** 2022-07-22

**Authors:** Marco Campisi, Sarah E. Shelton, Minyue Chen, Roger D. Kamm, David A. Barbie, Erik H. Knelson

**Affiliations:** 1Department of Medical Oncology, Dana-Farber Cancer Institute, Boston, MA 02115, USA; marco_campisi@dfci.harvard.edu (M.C.); sarahe_shelton@dfci.harvard.edu (S.E.S.); mchen@hms.harvard.edu (M.C.); david_barbie@dfci.harvard.edu (D.A.B.); 2Department of Biological Engineering, Massachusetts Institute of Technology, Cambridge, MA 02139, USA; rdkamm@mit.edu; 3Department of Immunology, Harvard Medical School, Boston, MA 02115, USA; 4Department of Mechanical Engineering, Massachusetts Institute of Technology, Cambridge, MA 02139, USA

**Keywords:** microphysiological systems, cell therapy, immunotherapy, model systems, CAR T, CAR NK, microfluidics

## Abstract

**Simple Summary:**

Cell therapy has transformed oncology and drug development, yet better model systems are needed to recapitulate the tumor immune microenvironment (TIME). Microphysiological systems (MPS) can comprehensively model the human TIME, including immune cells, endothelial cells, fibroblasts, matrix, and cytokines. This review discusses current barriers to developing cell therapies for solid tumors from the perspective of MPS model design approaches. Overcoming current limitations in model systems and advancing MPS engineering will facilitate oncology drug development.

**Abstract:**

Cell therapies, including adoptive immune cell therapies and genetically engineered chimeric antigen receptor (CAR) T or NK cells, have shown promise in treating hematologic malignancies. Yet, immune cell infiltration and expansion has proven challenging in solid tumors due to immune cell exclusion and exhaustion and the presence of vascular barriers. Testing next-generation immune therapies remains challenging in animals, motivating sophisticated ex vivo models of human tumor biology and prognostic assays to predict treatment response in real-time while comprehensively recapitulating the human tumor immune microenvironment (TIME). This review examines current strategies for testing cell-based cancer immunotherapies using ex vivo microphysiological systems and microfluidic technologies. Insights into the multicellular interactions of the TIME will identify novel therapeutic strategies to help patients whose tumors are refractory or resistant to current immunotherapies. Altogether, these microphysiological systems (MPS) have the capability to predict therapeutic vulnerabilities and biological barriers while studying immune cell infiltration and killing in a more physiologically relevant context, thereby providing important insights into fundamental biologic mechanisms to expand our understanding of and treatments for currently incurable malignancies.

## 1. Introduction: Promise and Challenges of Cell Therapies

Genetically modified immune cells, such as autologous or allogeneic CAR T and NK cells, represent a paradigm shift in oncology and cancer therapeutics (Figure 1). CAR T cells have been approved by the US Food and Drug Administration (FDA) for clinical use in hematologic malignancies including multiple myeloma, diffuse large B-cell lymphoma (DLBCL), mantle cell lymphoma, follicular lymphoma, and primary mediastinal B-cell lymphoma. Response rates in early trials were staggering, with complete responses in 54% of patients with DLBCL [1] and subsequent studies showed durable remissions in up to 40% of patients, including those with refractory disease [2]. These positive results have prompted the rapid development of CAR constructs targeting antigens beyond CD19 and BCMA for indications other than myeloma and lymphoma (Figure 1). 

As previously reviewed [3], results from trials of CAR T cells in solid tumors have been disappointing to date. A notable exception are the recent results from early-phase trials of claudin 6 and claudin 18.2 CAR T cells [4,5]. Proposed factors for decreased efficacy of CAR T cells in solid tumors include antigen specificity/expression, the dense/immunosuppressive tumor microenvironment, and the on-target/off-tumor toxicity [6]. In contrast to hematologic malignancies, solid tumors exhibit a more sophisticated TIME including cancer cells, infiltrated immune cells, fibroblasts, stromal cells, tumor vasculature, and other noncellular components [7] (Figure 2). This complex TIME diminishes CAR T and NK cell infiltration and homing. CAR engineering can overcome these challenges by targeting specific characteristics of TIME, such as the metabolic profiles of the CAR T or NK cells. For example, Juillerat et al. developed a novel CAR construct combining an oxygen-sensitive subdomain of HIF1α with a CAR scaffold; therefore, it can respond specifically to the hypoxic TIME in certain solid tumors [8]. Strategies targeting other TIME components, such as regulatory T cells or myeloid-derived suppressor cells, can also be used as combinational therapy along with CAR T or NK cells to modify the TIME and improve cell therapy efficacy [9,10,11,12,13,14]. 

There is hope that refined CAR engineering can improve transduced T and NK cell homing and efficacy (Figure 2), and many trials (>100 listed on ClinicalTrials.gov) with next-generation CAR T therapies are ongoing. NK cell therapies are entering the clinic and may offer advantages over engineered T cells including an improved safety profile and “off-the-shelf” manufacturing capability [7,8,11,13]. 

One challenge in developing cell therapies remains the lack of suitable model systems [13,14]. In vitro experiments can assess CAR potency and specificity but fail to model the TIME. Immunocompromised xenograft models can provide proof of principle for target engagement but likewise lack the complexity of the TIME. Syngeneic systems rely on mouse immunology and humanized mouse models bring caveats from engraftment that can limit translation. Microphysiological systems (MPS) and microfluidic technologies allow for the growth and treatment of human tumor cells with intact or recapitulated TIME, providing an opportunity to identify new therapeutic vulnerabilities and overcome current CAR therapy limitations [15]. Many of these models also offer precise control over geometry, cellular components, and physiochemical properties to help refine cell therapy engineering and understand barriers to efficacy in solid tumors.

## 2. Microphysiological Models for Cell Therapy Testing

Advanced preclinical models such as MPS have the potential to transform the drug development pipeline by predicting the outcome of clinical trials better than current animal models and traditional two-dimensional (2D) culture [15]. Preclinical models of human disease are essential for the basic understanding of disease pathology and development of efficient treatments for patients [16]. Nevertheless, most preclinical models have limitations in faithfully recapitulating the local tissue and organ microenvironment and, in certain circumstances, produce misleading outcomes [17]. For instance, static 2D cell culture models lack the complex multicellular interactions typical of in vivo tissues. Moreover, the gap between human 2D cell culture and animal models can dramatically affect clinical outcome [18,19] (Figure 3a,b). There are few existing models that can replicate continuous interactions and chemokine signaling between cells in the tumor microenvironment (TME) while also evaluating the preclinical efficacy of novel and personalized cancer therapeutics. Effective prediction of clinical outcomes is needed during preclinical testing of drug candidates to reduce high attrition rates in drug development [20,21]. In this context, engineered living systems using microphysiological technology represent one of the future platforms for in vitro experimentation and translational research, such as the testing of cell therapies (CAR T, CAR NK), innovative compounds and nanocarriers, and improving the reliability of models that mimic a broad spectrum of pathologies [14,21,22].

MPS platforms contain living cells that have been organized into compartmentalized single or multiple channels. The main goal of MPS modeling is to reproduce minimal functional units that recapitulate tissue- and organ-level functions in a more physiologically relevant model. Fundamental aspects of the technology are: (i) the selection of appropriate cell types, able to undergo self-organization into organ- or tissue-appropriate complex architectures and (ii) the design of appropriate extracellular matrix (ECM) composition that enables cell polarization and development through complex morphogenic steps, converging into precise anatomical structures. MPS use different approaches to generate 3D multicellular architecture in vitro. Broadly speaking, these methods can be categorized as spherical organoid and droplet models, bioprinting, and microfluidic systems [22]. Hence, the design of useful MPS should guide the process of cell self-organization into 3D multicellular structures [23,24]. 

### 2.1. Selection of Cell Types

Cell sources for MPS include fully differentiated primary cells and cell lines or stem cells, including both embryonic stem cells (ESC) and induced pluripotent stem cells (iPSC). ESC and iPSC have the capacity for spontaneous self-organization, reproducing pluripotent stem cell arrangement during embryogenesis to form in vitro “organoids” [24,25,26,27]. Indeed, in microphysiological systems, ESC and iPSC achieve higher expression of tissue-specific markers, suggesting more physiological behavior of stem cells in these models. However, the tumor spheroids most commonly used for MPS oncology studies are aggregates of cancer cell lines, either in monoculture or in combination with other cell types such as cancer-associated fibroblasts (CAFs) or endothelial cells (ECs), and in many instances lacking extracellular matrix. Immune components are typically incorporated with the addition of lymphocytes from peripheral blood mononuclear cells (PBMC), tumor infiltrating lymphocytes, CAR T, or CAR NK cells [28,29,30,31,32]. A unique approach to replicating the TIME in MPS is to collect organotypic spheroids directly from tumor tissues. These organotypic spheroids are small tumor fragments from human or mouse tissue generated by dissociating tissue specimens into fragments, typically ~100 μm in diameter. Organotypic tumor spheroids retain the native tumor stroma as well as many cell types, and lymphocyte populations can be maintained by supplementing IL-2 to sustain their growth within the spheroids or by adding PBMC or tumor-infiltrating lymphocytes derived from the same sample [32,33,34,35,36,37,38]. 

### 2.2. MPS Engineering with ECM

Either cell line spheroids or organotypic spheroids are suspended in hydrogel and injected and subsequently grown in one compartment of a microfluidic device. In yet another approach, cancer cells can be suspended within fluid drops to allow the mixing of multiple cell types in small volumes of fluid instead of encapsulation in hydrogel. For example, both lymphocytes and tumor spheroids can be combined inside hanging drops, such as a study which used this technique to assess the cytotoxicity of CAR T cells recognizing the human epidermal growth factor receptor 2 (HER2) in breast cancer [35,39]. Oil and water immersion microfluidic techniques can also be used to combine cancer cells with lymphocytes within droplets in order to image cytotoxicity or perform small-volume cytokine sampling [39,40,41,42,43,44,45,46,47,48]. 

Bioprinting can also be used to generate 3D cancer models, with more controllable architecture than organoid models, but generally lacking the dense stromal structure and heterogeneity of a patient tumor [45]. Many bioprinting approaches exist, such as inkjet, extrusion, acoustic, and laser photopolymerization methods [48]. For cell therapy, bioprinting has been used to improve T cell expansion and function using alginate and alginate–gelatin scaffolds to mimic lymph vessels, resulting in the differentiation of CD4+ cells into the central memory type and differentiation of CD8+ cells into effector memory type [45]. Bioprinted tumor spheroids have been used to test infiltration and efficacy of novel CAR T cells, such as a study in neuroblastoma testing CAR T cells recognizing the L1 cell adhesion molecule [47]. The controllable geometry made possible by bioprinting also allows for the creation of multiple structures, such as models that print both large-scale vascular channels and tumor spheroids [48]. 

Microfluidic devices for MPS containing adjacent chambers and channels for cell/hydrogel or fluidic compartments can be fabricated from a variety of materials, but often rely on high-precision soft lithographic methods with polydimethylsiloxane (PDMS), a transparent, inert, and biocompatible polymeric substrate. Soft lithography or other methods can be used to create features at the micron scale in configurations that typically enable live microscopy, addition and sampling of fluids, and controllable gradients across devices [49,50]. Microengineered devices, by providing spatial boundary conditions and control over these parameters, can direct these processes, and therefore have proven to be well-suited to guide complex tissue organization [51,52].

The wide variety of approaches possible for generating MPS allows researchers to use simple systems to facilitate 3D in vitro oncology studies or to design custom platforms that enable unique tissue-on-a-chip configurations or new quantification methods and opportunities for novel hypothesis testing. MPS studies in oncology and cell therapy have been accelerating, likely due to several advantages they possess over traditional 2D cell culture and in vivo studies using small animal models.

## 3. Advantages of Microphysiological Systems (MPS)

MPS offer several advantages, especially for cell therapy studies in oncology. These multicellular, 3D culture platforms can be designed to resemble the complex architecture of living tissue more closely, and multiple cell types can be combined in culture systems accessible to high-resolution imaging, fluid sampling, and other quantification techniques for monitoring interactions between cells. Furthermore, many of these models also incorporate extracellular matrix, which has been observed to influence cell phenotype, immune cell migration, and cytotoxicity efficiency. Finally, these models can make use of cells from any organism, making it possible to generate fully human, 3D, in vitro systems closely mimicking tissue architecture and overcoming the disadvantages of small animal models for immune-oncology studies (Figure 4). 

Though it is common in MPS oncology studies to use a combination of cancer cells and immune cells in 3D, they vary widely in how the cells are organized or introduced. The simplest models are tumor spheroids, either from established cell lines or patient-derived sources, with a suspension of immune cells added. These models typically place tumor spheroids in microwells to image the infiltration and cytotoxicity of lymphocytes [41,53,54,55,56,57,58,59,60]. Other studies encapsulate tumor spheroids in hydrogels such as collagen or Matrigel, either embedding the lymphocytes or applying them to the surface of the hydrogel [29,48,58,59,60,61,62,63,64,65,66,67]. The advantage of performing these studies in 3D is that lymphocytes must actively migrate through the ECM toward and into the tumor spheroids in order to exert cytotoxic effects, rather than the passive interactions that occur in traditional cytotoxicity assays in which lymphocytes settle onto a 2D cancer cell monolayer. In a study using TCR-engineered T cells recognizing HLA-A0201 and a hepatitis B epitope, they found that after 15 h, there was nearly complete killing of hepatocellular carcinoma observed in 2D, but only 25% killing achieved with the same conditions arranged in 3D [60]. In addition to the limited access presented by the 3D tumor setting, the authors attributed the difference to the lack of increased HLA expression in response to inflammatory cytokines in 3D, similar to an observation made by another group comparing T cell recognition in melanoma [62]. NK cells were also less effective in 3D compared to 2D configurations, especially at lower effector to target ratios (E:T) [58], and this could be due to the inhibitory effect on NK cells of increased HLA-E expression that occurred in 3D [58].

Cytotoxicity efficiency can be reduced by nutrient deficiencies and metabolic changes that occur in larger 3D systems. For example, Ayuso et al. used a rectangular microfluidic system that was supplied with cell culture media from only one side to form a gradient across the width of the device [58]. They found uneven distribution and cytotoxic efficiency across the device and attributed the differences in NK gene signatures in the proximal vs. distal regions to nutrient deprivation rather than hypoxia. They suggested that NK cells that travelled the length of the device became less proliferative, less responsive to chemokines, and more proinflammatory. Even after liberating the NK cells from the device and returning them to flask culture, these cells were less effective at killing than fresh NK cells and alterations in prosurvival genes remained, suggesting a “stress memory”. Hypoxia also reduced migration, infiltration, and cytotoxicity of TCR T and CAR T cells [64,66]. 

Microfluidic systems are also ideal for mimicking flow conditions in and around the tumor, serving several useful purposes including the delivery of immune cells that are suspended in the flowing medium and allowed to contact and adhere to either the gel surface or other cells in order to understand lymphocyte adhesion under shear stress. Rosa et al. cultured a monolayer of dendritic cells and designed a microfluidic device to direct streams of CD4+ and CD8+ cells across it at defined shear stress levels [65]. T cells were able to adhere under low shear stress but detached when shear stress reached 12 Dyn/cm^2^. Another study tracked the speed of interactions between T cells under flow and immobilized beads coated with MHC to infer the affinity of the receptor-ligand interactions [66]. A different approach to flow is connecting multiple tissue compartments with fluidic channels, and one such study observed the impact of paracrine factors resulting from NK elimination of colorectal cancer spheroids on cardiac tissue organoids connected by fluidic channels [67]. They found that the viability and morphology of the cardiac spheroids were unchanged, but the beat frequency decreased, and the amplitude became less stable over 1 week of coculture. 

Microphysiological platforms that support the culture of multiple cell types can, therefore, also be used to investigate the roles of additional cells making up the stroma of the tumor microenvironment, such as ECs (vascular or lymphatic), fibroblasts, and other immune cell types such as antigen-presenting cells and myeloid-derived suppressor cells. CAFs can exert immunosuppressive functions in multiple ways, primarily through remodeling of the extracellular matrix and secretion of paracrine factors that inhibit lymphocyte function. CAF are often combined directly with tumor spheroids or embedded in the gel surrounding them [55,68,69,70,71,72,73,74,75,76,77]. When peripheral blood mononuclear cells (PBMC) were co-embedded with cancer cells and CAF, Nguyen et al. found that including CAF in a model of antibody-dependent cell-mediated cytotoxicity (ADCC) reduced the efficacy of the drug trastuzumab [71]. 

In these tissue models where cells are embedded in hydrogels, not only are lymphocytes required to navigate through the gel to reach their targets, but they are also subjected to the mechanical properties of the substrate, such as density and stiffness. Lymphocytes exert their killing functions through direct contact with target cells, and as such are exquisitely sensitive to mechanical stimuli and use these signals to guide their migration and activation [72,73]. While increasing ECM density (in collagen) decreases the migration speed of both NK cells [58] and T cells [74], increased stiffness enhances lymphocyte cytotoxicity [60,75,76]. T cells sense the stiffness of their environment and can respond with increased proliferation, migration, and activation [76]. NK cells also demonstrated more rapid killing in higher density collagen gels [54]. The enhanced killing efficiency seen in stiff matrices is due, in part, to increased membrane tension [77,78]. Perforin-mediated cell death is more rapid and effective in target cells with higher membrane tension imposed by stiffer substrates, a finding which was also confirmed through the use of hyper- and hypotonic solutions, blebbistatin, and latrunculin to modify cancer cell membrane tension in B16 murine melanoma [75]. 

ECs also play an important role in the lymphocyte infiltration of solid tumors, since NK and T cells traveling through circulation must bind to the endothelium and extravasate before migrating toward tumors. While some models incorporate an endothelial monolayer to enforce the binding/extravasation step of lymphocyte homing to tumors [64,73,74], other models strive to form cylindrical, micropatterned, or other vascular geometry that is more representative of capillary beds [58,78,79].

## 4. Microphysiological Systems including Microvasculature 

Tumors are complex ‘ecosystems’ comprised of many different cell types and noncellular features, and it has been demonstrated that the complexity and diversity of the TIME influences patient response to immunotherapy. The tumor stroma, including ECs, pericytes, CAFs, etc., has a critical role in tumor initiation, progression, metastasis, and therapy resistance [80,81]. In many tumors, the vasculature is abnormal, and a dysfunctional imbalance between the levels of proangiogenic and antiangiogenic factors promotes rapid but aberrant vascular formation. Morphologically, tumor blood vessels are tortuous, dilated, and unevenly distributed with impaired perfusion, thus exerting direct impact on cell function and viability [81]. Angiocrine cues participate actively in the induction, patterning, and guidance of organ formation and regeneration, as well as in the maintenance of homeostasis and metabolism [82]. ECs also supply stimulatory or inhibitory growth factors, produce ECM components, and collectively release cytokines that consequently affect the tissue-specific microenvironment, such as in the lung tumor microenvironment [79], and in the brain [82]. Though embedding cancer spheroids and stromal cells in the ECM and introducing lymphocytes from side channels allows researchers to observe lymphocyte migration and homing towards tumors [83,84,85,86,87,88,89,90,91,92,93,94,95,96,97,98,99], the addition of an endothelial barrier better recapitulates the process of extravasation [100,101,102,103]. Few models have also incorporated long-term tumor cell [99] or immune cells flow or recirculation [103], as this is a remaining challenge in the field. 

Modeling vascularization is key to cell therapy testing for a variety of reasons. First, vascularized models incorporate the angiocrine functions of ECs which interact with cancer cells to shape the TIME [81]. Second, these models enable the recapitulation of organ-specific barriers (such as the blood–brain barrier) or the tumor–vascular barrier impairing immune cell infiltration into solid tumor microenvironments. Finally, the long-term stability of in vitro models can be maintained and even increased by vascularization to improve nutrient delivery, gas exchange, and waste elimination. 

### 4.1. Engineering Vascularized MPS

In vitro models of blood vessels are typically generated by two distinct bioengineering approaches: via endothelial-lined patterned channels or by self-assembled networks [86,92,94,99]. Both systems are developed using microfluidic technology [49,50], with advantages such as providing gas exchange and nutrient delivery, enabling the study of extravasation of metastatic cancer cells [86,87,88,89] to reach a more physiologically-relevant model of the tissue-specific TIME. 

The patterned microchannel method consists of the 3D culture of cells on the walls of micropatterned chips or gel surfaces. Using this method, the channel design determines the size and architecture of the newly formed vessels. To develop hollow-lumen 3D macrovessels, ECs adhere to microfluidic channel walls or cylindrical casts, precoated with cell adhesive materials, such as fibronectin, Matrigel, or collagen [90,91,92,93,94,95] (Figure 5). Depending on the design of the microfluidic device, the 3D macrovessel has one or multiple sides that share borders with the hydrogel/cell channels. The size of the 3D macrovessel depends on the size of the fluidic channels, typically raging between 100 and 2000 µm [98]. These vessels are suitable for mimicking large blood vessel circulation, permeability, cell extravasation, and cell–cell interactions [84]. Macrovessels can also be used in combination with angiogenesis, with in vitro sprouting stimulated proangiogenic gradients created within the device through the addition of recombinant growth factors or produced by stromal cells in the device [95,96].

Alternatively, microvascular networks can be created by leveraging a vasculogenesis-like process that occurs when ECs are embedded in hydrogel in suitable conditions [94,95,96,97,98,99,100,101,102,103]. The vasculogenesis approach mimics early embryonic formation of the vascular plexus, during which blood vessels are formed by mesoderm-derived endothelial progenitor coalescence followed by the formation of hollow lumens and differentiation into mature ECs [86]. Expansion and sprouting from mature vasculature subsequently occurs through angiogenesis, such as occurs in tumors [88,89]. In vitro generation of self-assembled microvascular networks begins with the introduction of ECs (HUVEC, ECFC, or iPSC-EC) and stromal cells (fibroblasts or pericytes) suspended in conducive hydrogel such as fibrin gel, into a chamber of a microfluidic device. ECs connect and form lumens over 3–7 days, resulting in microvascular formation with a more physiological morphology and function. 

### 4.2. Perfusing Vascularized MPS

To provide effective function, it is important that the vascular networks be perfusable, as can be confirmed by introducing fluorescent tracers such as dextran or beads [83]. Self-assembled microvessels can reach capillary-scale diameters more easily than cast structures [104,105], but their morphology cannot be controlled due to their dependence on natural formation. The ability to generate perfusable microvascular networks in vitro has, however, allowed the visualization of physiologically relevant events in realtime, such as transendothelial extravasation of tumor cells [103,104,105], as well as the modeling of vascularized microtumors [87]. Additionally, vascular networks tend to be stabilized by paracrine signaling with stromal cells, as they regress soon after formation in their absence [83] (Figure 6).

## 5. Design of 3D MPS Using Microfluidic Technology

The key to success in developing any advanced preclinical model is to focus on mimicking organ-level physiology or pathophysiology observed in vivo. Validation depends on the demonstration that the model can effectively recapitulate behaviors observed in vivo [18,106]. Over the past decade, in addition to microfluidic technology, a wide variety of bioengineering approaches and strategies have been developed as alternatives to traditional cultures, including organoid technology and tissue engineering techniques, such as 3D bioprinting, as discussed above [15,107,108]. The great potential of MPS is to reproduce tissue-specific architecture; however, printing speed and resolution are important limitations. Similarly, it is possible to print 3D blood vessels, but at present these remain limited to vasculature with large diameters, and only a few such printing technologies exist, such as light-assisted bioprinting techniques, to achieve cell scale resolution approaching capillary sizes [104].

MPS using microfluidic technology have some of the characteristics of other advanced models (3D bioprinting and organoids) [25], such as dynamic 3D culture with complex microenvironment to mimic cell–cell interactions [15,20,109,110]. MPS have shown the ability to recapitulate key microenvironmental characteristics of human organs and mimic their primary functions [106]. Results obtained from microfluidic devices tend to be highly reproducible, relevant, and more directly translatable to humans. One of the main advantages of these platforms is the ability to control the specific cell and tissue architecture to emulate chemical gradients and biomechanical forces [16]. This allows for precise control over the biochemical and cellular milieu to model in vivo-like environments and responses [31,32,33]. Compared to other advanced 3D models, microfluidic culture allows the precise formation of microvascular networks and large blood vessels to mimic multicellular vascular interactions.

Many extracted immune cells have short life spans ex vivo, for example as little as 1 day for neutrophils, and on the order of a few weeks for lymphocytes. Since MPS require few cells, microfluidic culture can conserve immune cells, as seen in applications with patient-derived organotypic tumor spheroids (PDOTS) and murine MDOTS [33]. These platforms have been used for viability assays to evaluate drug toxicity and metabolism for current immunotherapy as well as mechanisms of drug resistance and the biology of drug-tolerant persister cells [108,109,110,111,112,113,114,115].

## 6. Translational Applications of MPS

MPS also offer the possibility of improving personalized medicine by more faithfully recapitulating patient-specific, organ-level pathophysiology and responses to therapies [108]. Indeed, an additional advantage gained by an even more comprehensive human patient-derived in vitro model, to generate patient-specific microphysiological systems at an integrative system level is critical for personalized medicine [109]. For example, one could combine iPSCs and/or patient-derived EC models. Advances in ex vivo modeling may allow long-term culture of patient-derived tumor samples to enable a personalized medicine approach to study drug resistance and immune cell penetration in patient-specific tumor niches [110]. Animal models, especially inbred research mice, are in widespread use, but there are limitations to using these models in immune-oncology studies. Patient-derived xenograft (PDX) models can capture many features of human tumor genetics and heterogeneity, but researchers must use immunodeficient mice to enable the successful growth of xenograft tissue. Humanized mice can recapitulate some aspects of the human immune system via the introduction of PBMC, tumor-infiltrating lymphocytes, or hematopoietic stem cells from the same patient [112]. However, this can induce a graft vs. host response within a few weeks, limiting the longevity of immunotherapy studies possible with these models. Furthermore, the establishment and maintenance of PDX colonies requires significant expertise, time, and cost. 

MPS systems offer the alternative of generating fully human 3D systems that can be used to culture either established cell lines or patient-derived tissues. Patient-derived organotypic tumor spheroids (PDOTS) have been embedded in microfluidic devices to enable research groups to track immune cell migration towards spheroids, monitor cell death with live/dead stains, and to perform immunostaining to identify different cellular populations within the samples, which can come from biopsies or surgical resections [31,32,33,34,35,36,37]. Furthermore, ex vivo patient-derived tumor spheroids retain drug sensitivities of the primary tumor, similar to what is seen in PDX models as well [34,35,36]. These capabilities create opportunities to use MPS in conjunction with experimental cell therapies in precision clinical trials [113,114,115,116,117,118,119,120,121].

## 7. Predicting Therapeutic Vulnerabilities and Biologic Barriers Using MPS

One of the obstacles to immune cell homing is the vasculature barrier facing CAR T or NK cells injected intravenously. Only a small portion of these cells can extravasate through the vasculature barrier and enter the TIME to kill tumor cells. Hence, strategies to improve the extravasation and migration of CAR T cells are critical for improving cell therapy efficacy. We previously showed that the stimulator of interferon genes (STING) plays a crucial role in antitumor immune responses and restoring or activating cGAS-STING signaling in both tumor cells and ECs promotes immune cell recruitment via the CXCR3-CXCL10 receptor-chemokine axis and vascular activation [116]. As discussed above, upregulation of adhesion molecules on the vasculature can help immune cell extravasation and migration into the TIME, improving cell therapy effectiveness. Another physical barrier to immune cell extravasation is the dense stroma. This is exemplified by the regional administration of CAR T or NK cells, which are better able to physically cross the stromal barrier, enhancing efficacy [117]. Another approach is to engineer CAR T or NK cells to express heparinase, an ECM-degrading enzyme that can reduce the stromal barrier to increase the potential for immune cell trafficking [118].

## 8. Remaining Challenges in Modeling Response to Cell Therapies 

Despite innovations in terms of complexity, in vitro devices are still unable to fully recapitulate biological in vivo interactions within an organism. With recent advances in fluidically coupled multiple microfluidic systems to connect microphysiological systems and recreate a human body-on-chip, we can begin to consider the possibility of creating multi-MPS models to mimic the systemic interactions between organs [119]. One opportunity is to model systemic metastasis using a multi-organ-on-chip or body-on-chip system. The metastatic spread might be modeled from a vascularized lung model to the brain, passing through either the blood vasculature or lymphatic network via lymph nodes [20]. These systems will still be simplistic representations of organ interactions, far from in vivo models in mice or humanized mice [120,121], but with the advantages of using exclusively human cells and enabling observation and control in real-time. In light of the biochemical and structural complexity of organs, it is unlikely that a single engineering approach for building artificial ex vivo tissues will solve the challenges of manufacturing and modeling artificial equivalents of human biology for preclinical systems [24,122].

One of the disadvantages of microfluidic platforms is that they are difficult to maintain for long-term culture (months), and experimental protocols involving microfluidic devices require experienced operators. Microfluidic devices, compared to multiwell plates, could be considered as low-throughput screening tools (but potentially medium to high throughput screening). One possibility will be to design more complex platforms containing an increased number of devices. Additionally, automation will help increase throughput [123]. A key improvement to the current models is the introduction of continuous perfusion by integrating micropumps [124]. Indeed, flow perfusion culture could advance the model in several important aspects. First, it enhances transport of nutrients and gas exchange for maintenance of long-term cultures. Second, flow-mediated shear stress improves microvascular formation and reduces vascular permeability [101,125]. Specifically, flow stimuli have been demonstrated to promote the differentiation of vascular ECs into a more physiological phenotype, with the highest expression of tight junction proteins and membrane transporters, producing further reductions in permeability [126]. Lastly, perfusion can enable the introduction of circulating immune cells and model the effects of fluid shear stress on the arrest and adhesion of immune cells. 

Inclusion of additional cells, such as lung epithelial cells for the lung models and lymphatic vessels and fibroblasts to generate lymph nodes, will create a more faithful recapitulation of the TIME [127,128]. However, this is currently limited to the use of cell lines or stem cells. One of the main hurdles to the development of patient-specific microphysiological cancer models that enable the sophisticated level of analysis described above is the need to isolate cell types (cancer cells, ECs, stromal cells, and immune cells) from the same patient. One could then construct organ chips with the appropriate cell types in the correct relative proportions and location to accurately mimic in vivo behaviors and responses [128]. The challenge of this approach is that each cell type requires specific isolation protocols and culture conditions to optimize cell-specific functionality. Furthermore, meaningful integration of multiple cell types in microfluidic devices can be time-consuming and requires extensive optimization [21,129]. Furthermore, histocompatibility becomes an issue when combining patient-derived organotypic tumor spheroids with mismatched ECs to generate vascularized tumor models. To avoid alloreactions, it will be necessary to obtain primary ECs, ECFC, or generate iPSC-ECs from the same patient for personalized medicine applications [130].

## 9. Microphysiological Systems in Pharmaceutical Development

MPS have shown promising results and value, but the field is in its infancy in terms of preclinical application and standard inclusion in drug discovery pipelines. MPS are not yet robust enough to support routine and widespread adoption, but could eventually be incorporated into existing screens, traditional workflows, and platforms. In support of their adoption, MPS were used to test >100 compounds for which clinical data were available [130]. It will be necessary to systematically compare these MPS models with traditional platforms and animal models and define standardized methods and protocols to compare systems. Minimum criteria for acceptance of MPS include the ability to reproduce the desired physiology and functionality and elicit the expected responses to standard or reference compounds and assays, while also demonstrating reproducibility and reliability at scale [131]. To realize the full potential of MPS, more collaboration will be needed between stakeholders: regulators, pharmaceutical companies, and academic investigators. A recent survey forecast that within 5 years, MPS would save between 10% and 26% of R&D costs, with the greatest impact being realized during the lead optimization process [132]. However, before this can happen, companies need to make an upfront investment in the technology, as most of the approaches are still exploratory and require substantial refinement before use in a drug development setting. To this end, several international consortia including the Innovation and Quality (IQ) consortium [133], the European society for alternatives for animal testing (EUSAAT) [134], the European Organ on chip society (EUROoCS) [135], and the Johns Hopkins University Center for Alternatives to Animal Testing (CAAT) [136], are working to promote advanced preclinical models including MPS. In summary, bridging the gap between proof of concept and ‘industry-grade’ MPS for testing drugs and nanocarriers requires investment, collaboration, and communication from all stakeholders. This innovative field, at the intersection of biology and engineering, has an unprecedented opportunity to advance preclinical models. Future work requires systematic validation, navigating regulation by government entities, reframing the challenge, and adapting traditional biomedical research practices to develop new approaches for drug discovery and development.

## 10. Conclusions

In contrast to hematologic malignancies, the major limitation to cell therapies for solid tumors is the necessity for lymphocytes to infiltrate tumor tissue and remain active in the presence of immunosuppressive signals supplied by the TIME. The barriers to effective lymphocyte homing are imposed by the tumor architecture, including vascular barriers to extravasation, cytokine gradients to guide migration, and mechanical barriers imposed by the extracellular matrix. Once lymphocytes reach the tumor tissue, they often encounter a highly immunosuppressive environment generated by the tumor and stroma, such as metabolic conditions, paracrine signaling, and checkpoint inhibition. Together, this complex TIME results in the exclusion or exhaustion of cell therapies without a clear indication of which feature is the major limitation. Therefore, MPS offer an ideal approach to identify the mechanisms underlying these barriers, allowing researchers to design novel strategies for overcoming them. The many advantages of 3D, multicellular, human MPS allow them to replicate cancer pathophysiology more fully than traditional cell culture methods. With regard to long-term impact, refining and expanding the repertoire of MPS models in oncology will enable improvements in the testing of emerging cell therapies and facilitate the design of new drugs to overcome the barriers of the TIME. Furthermore, patient derived MPS enable stratification and identification of individual responses to therapy. These numerous advantages will result in an increasing number of approaches to MPS tailored for cell therapy studies in the future, leading to continued improvements in patient outcomes. 

## Figures and Tables

**Figure 1 cancers-14-03561-f001:**
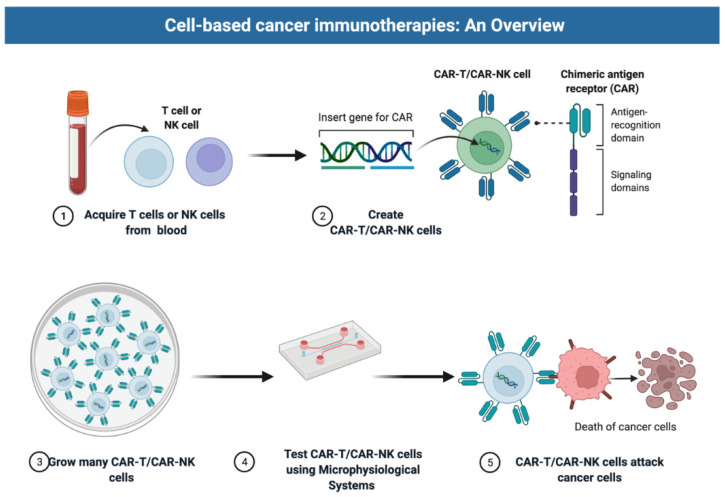
Overview of cell-based immunotherapies. The manufacturing process of CAR T and CAR NK cell therapies starts with the isolation of PBMCs, collected either from whole blood phlebotomy or, more commonly, through a leukapheresis procedure. The selection or depletion of specific T-cell or NK-cell types within the PBMCs is then performed. In some protocols, the cells are enriched for CD3+ T cells prior to or concurrent with their activation, at which point the cells are genetically modified using viral vectors or other nonviral gene delivery methods to express the CAR. The cells are then expanded in the presence of cytokines. The gene-modified cells can be added to microphysiological systems to test their cytotoxic activity. Schematics created with BioRender.com (accessed on 25 May 2022).

**Figure 2 cancers-14-03561-f002:**
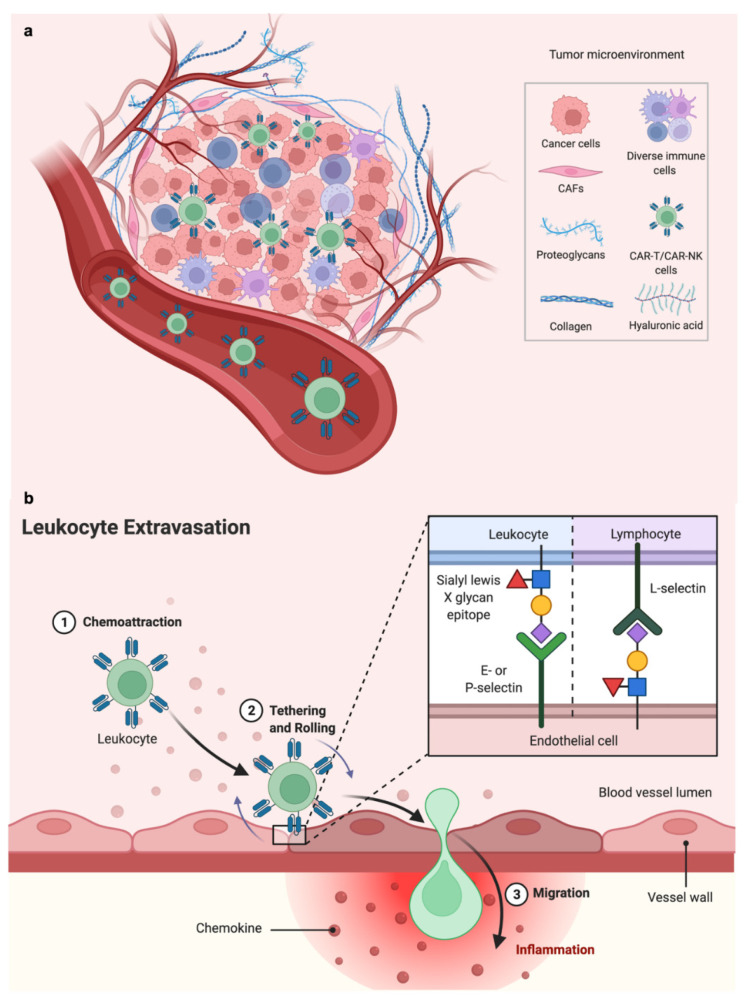
Overview of the tumor microenvironment and lymphocyte homing. (**a**) Components of the tumor immune microenvironment (TIME). The TIME is a complex ecosystem of heterogeneous tumor cells, stromal cells, and immune cells residing in a network of dysregulated vasculature and extracellular matrix (ECM) proteins. CAR T and CAR NK cells traffic through the vascular and stromal barrier to reach the TIME. (**b**) Leukocytes are recruited from the bloodstream to cancer tissues via a sequential multistep process known as the leukocyte adhesion cascade or extravasation. First, endothelial activation is triggered by stimuli from the inflamed tissue to produce chemokines, which triggers selectin-dependent tethering and rolling of leukocytes along the luminal surface of the blood vessel. Subsequently, chemokines are presented on the luminal surface of the endothelium, which activates leukocyte-expressed integrins allowing bond formation with their endothelial-expressed ligands, resulting in leukocyte arrest followed by transendothelial migration. (Schematics created with BioRender.com accessed on 25 May 2022).

**Figure 3 cancers-14-03561-f003:**
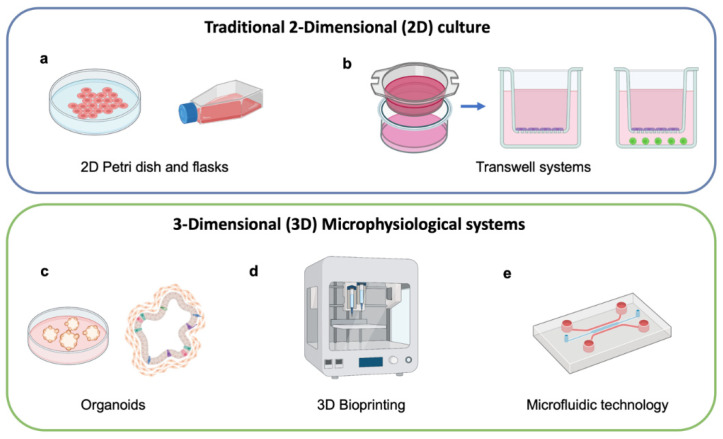
Traditional 2-dimensional culture and 3-dimensional microphysiological systems. (**a**) 2D culture system of cell lines, (**b**) transwell systems containing monoculture or co-culture, (**c**) organoids, (**d**) 3D bioprinting system, and (**e**) microphysiological system using microfluidic technology. (Schematics created with BioRender.com accessed on 26 May 2022).

**Figure 4 cancers-14-03561-f004:**
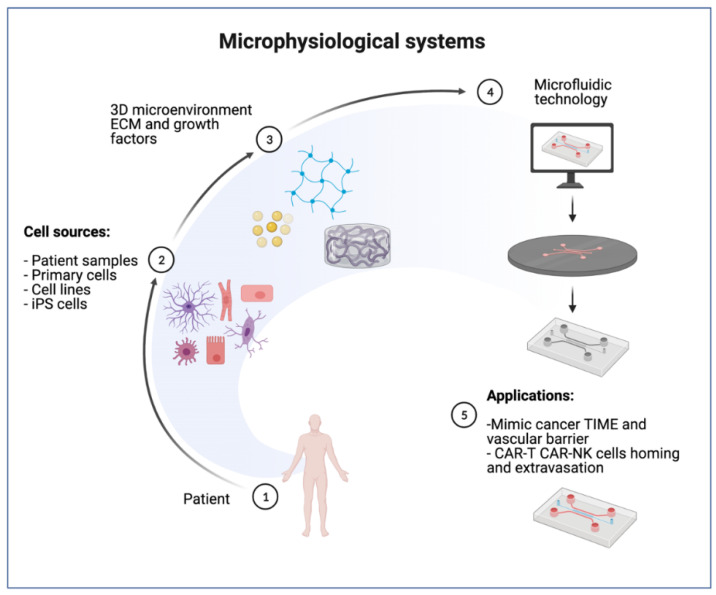
3D microphysiological systems using microfluidic technology. Schematic representation of microphysiological system: cells sources, extracellular matrix (ECM) growth factors, and fabrication processes of microfluidic technology. Devices are loaded with cell combinations including CAR T or CAR NK cells for several applications including mimicking the tumor immune microenvironment and vascular barrier to model immune cell homing and extravasation. (Schematics created with BioRender.com accessed on 27 May 2022).

**Figure 5 cancers-14-03561-f005:**
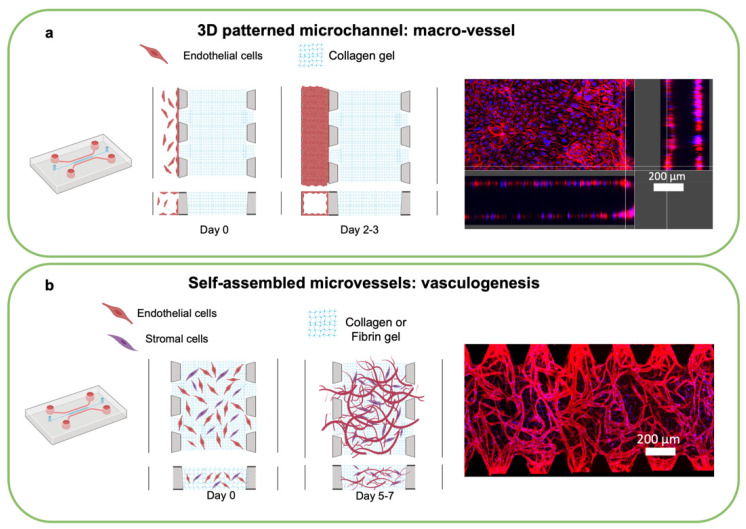
Engineering vascular formation in microphysiological systems. (**a**) Schematic of a patterned microchannel: macrovessel formation in the microfluidic device using endothelial cells (ECs) over 2–3 days. Confocal image of F-actin (red, ECs) and DAPI (blue, nuclei) at day 3 adapted from [32]. (**b**) Self-assembled microvessels by vasculogenesis. Schematic of the microvascular network formation after 7 days. Cells (ECs and stromal cells) self-organized into microvasculature. Confocal image of F-actin (red, ECs) and DAPI (blue, nuclei) at day 7. Adapted from [101] with permission. Schematics created with BioRender.com accessed on 27 May 2022.

**Figure 6 cancers-14-03561-f006:**
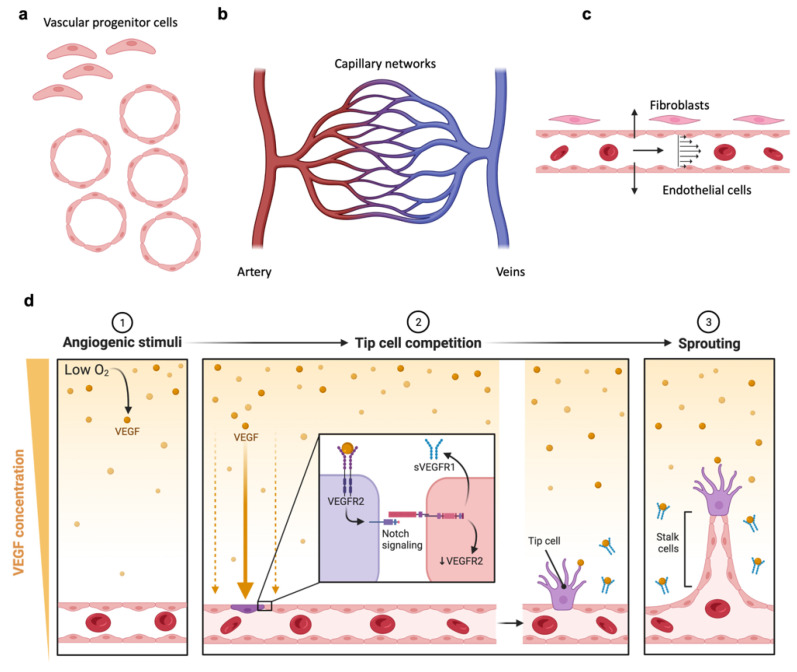
Vascular network formation in vivo. (**a**) Vascular progenitor cells merge into endothelial cells islands, which subsequently form a microvasculature. (**b**) Schematic of connected arterioles and venules after vasculogenesis. (**c**) Section of a blood vessel showing mechanical factors such as wall shear stress and axial strain, also direct angiogenesis. (**d**) EC-secreted factors recruit mural cells during angiogenic remodeling. Proangiogenic factors are released from tumor cells and stromal cells such as fibroblasts directing migration and sprouting of ECs. (Schematics created with BioRender.com accessed on 25 May 2022).

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
