# Peer review of "Engineered Microphysiological Systems for Testing Effectiveness of Cell-Based Cancer Immunotherapies"

_cancers, 2022, doi:10.3390/cancers14153561_

Round 1

Reviewer 1 Report

The manuscript by Marco Campisi and co-authors review the creation of in vitro preclinical models that accurately reproduce human solid tumors. The text is well structured and illustrated. The article may be of interest to a wide audience of scientists.

Minor remarks:
Perhaps it makes sense to divide some sections (for example, 2 and 4) into subsections for better understanding of the text.
Figure 5 appears to have a missed collagen gel icon (blue grid).

It would be also interesting to see an examples (in form of some table) of the use of MPS in pre-clinical development that translates into clinical trials.

Author Response

The manuscript by Marco Campisi and co-authors reviews the creation of in vitro preclinical models that accurately reproduce human solid tumors. The text is well structured and illustrated. The article may be of interest to a wide audience of scientists.

-Thank you for the positive feedback!

Minor remarks:
Perhaps it makes sense to divide some sections (for example, 2 and 4) into subsections for better understanding of the text.

-We appreciate this suggestion and have added sub-sections to sections 2 and 4 to enhance organization and clarity.

Figure 5 appears to have a missed collagen gel icon (blue grid).

-Thank you for catching this—we have updated Figure 5.

It would be also interesting to see an examples (in form of some table) of the use of MPS in pre-clinical development that translates into clinical trials.

-This is a great idea and we would love to generate this type of table but worry it may not be possible. MPS have only recently been used in pre-clinical development, without clear timeliness and disclosures by companies on how preclinical experiments are being used to inform clinical trials. We share the reviewer’s hope that MPS will become a standard part of preclinical development, allowing clear comparisons of ex vivo results with trial data.

Reviewer 2 Report

This review is very well written. I don't think that there is need to revise. 

Author Response

Thank you for the positive feedback!

Reviewer 3 Report

Campisi et al in their review discuss about the recent advancements and limitations of microphysiological systems in novel cell-based cancer immunotherapies.

Genetically engineered CAR (or NK) T cells are promising tools in cancer therapy nowadays. However, this novel technique bears numerous barriers, thus development of new in vitro methods are important to optimize and monitor this cancer treatment strategy.

The manuscript is well-written, easy to follow and interpret.

All figures are well-constructed and clear and help the reader excellently to describe the tumor immune microenvironment and various 2D/3D microphysiological systems.

Summarizing the latest developments in mimicking of the tumor immune microenvironment is timely and highly important.

I recommend the acceptance of the manuscript in its current form.

Author Response

Thank you for the positive feedback!